# Hyperspherical Prototype Networks

**Pascal Mettes**
ISIS Lab
University of Amsterdam

**Elise van der Pol**
UvA-Bosch Delta Lab
University of Amsterdam

**Cees G. M. Snoek**
ISIS Lab
University of Amsterdam

## Abstract

This paper introduces hyperspherical prototype networks, which unify classification and regression with prototypes on hyperspherical output spaces. For classification, a common approach is to define prototypes as the mean output vector over training examples per class. Here, we propose to use hyperspheres as output spaces, with class prototypes defined *a priori* with large margin separation. We position prototypes through data-independent optimization, with an extension to incorporate priors from class semantics. By doing so, we do not require any prototype updating, we can handle any training size, and the output dimensionality is no longer constrained to the number of classes. Furthermore, we generalize to regression, by optimizing outputs as an interpolation between two prototypes on the hypersphere. Since both tasks are now defined by the same loss function, they can be jointly trained for multi-task problems. Experimentally, we show the benefit of hyperspherical prototype networks for classification, regression, and their combination over other prototype methods, softmax cross-entropy, and mean squared error approaches.

## 1 Introduction

This paper introduces a class of deep networks that employ hyperspheres as output spaces with an *a priori* defined organization. Standard classification (with softmax cross-entropy) and regression (with squared loss) are effective, but are trained in a fully parametric manner, ignoring known inductive biases, such as large margin separation, simplicity, and knowledge about source data [28]. Moreover, they require output spaces with a fixed output size, either equal to the number of classes (classification) or a single dimension (regression). We propose networks with output spaces that incorporate inductive biases prior to learning and have the flexibility to handle any output dimensionality, using a loss function that is identical for classification and regression.

Our approach is similar in spirit to recent prototype-based networks for classification, which employ a metric output space and divide the space into Voronoi cells around a prototype per class, defined as the mean location of the training examples [11, 12, 17, 37, 45]. While intuitive, this definition alters the true prototype location with each mini-batch update, meaning it requires constant re-estimation. As such, current solutions either employ coarse prototype approximations [11, 12] or are limited to few-shot settings [4, 37]. In this paper, we propose an alternative prototype definition.

For classification, our notion is simple: when relying on hyperspherical output spaces, prototypes do not need to be inferred from data. We incorporate large margin separation and simplicity from the start by placing prototypes as uniformly as possible on the hypershere, see Fig. 1a. However, obtaining a uniform distribution for an arbitrary number of prototypes and output dimensions is an open mathematical problem [31, 39]. As an approximation, we rely on a differentiable loss function and optimization to distribute prototypes as uniformly as possible. We furthermore extend the optimization to incorporate privileged information about classes to obtain output spaces with semantic class structures. Training and inference is achieved through cosine similarities between examples and their fixed class prototypes.

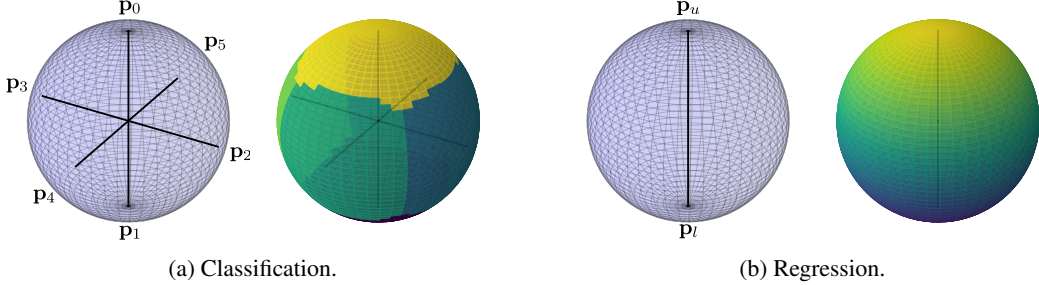

(a) Classification.                                    (b) Regression.

Figure 1: This paper demonstrates that for (a) classification and (b) regression, output spaces do not need to be learned. We define them *a priori* by employing hyperspherical output spaces. For classification the prototypes discretize the output space uniformly and for regression the prototypes enable a smooth transition between regression bounds. This results in effective deep networks with flexible output spaces, integrated inductive biases, and the ability to optimize both tasks in the same output space without the need for further tuning.

Prototypes that are *a priori* positioned on hyperspherical outputs also allow for regression by maintaining two prototypes as the regression bounds. The idea is to perform optimization through a relative cosine similarity between the output predictions and the bounds, see Fig. 1b. This extends standard regression to higher-dimensional outputs, which provides additional degrees of freedom not possible with standard regression, while obtaining better results. Furthermore, since we optimize both tasks with a squared cosine similarity loss, classification and regression can be performed jointly in the same output space, without the need to weight the different tasks through hyperparameters. Experimentally, we show the benefit of hyperspherical prototype networks for classification, regression, and multi-task problems.

## 2    Hyperspherical prototypes

### 2.1    Classification

For classification, we are given $N$ training examples $\{(\mathbf{x}_i, y_i)\}_{i=1}^N$, where $\mathbf{x}_i \in \mathbb{R}^L$ and $y_i \in C$ denote the inputs and class labels of the $i^{th}$ training example, $C = \{1, .., K\}$ denotes the set of $K$ class labels, and $L$ denotes the input dimensionality. Prior to learning, the $D$-dimensional output space is subdivided approximately uniformly by prototypes $\mathbf{P} = \{\mathbf{p}_1, ..., \mathbf{p}_K\}$, where each prototype $\mathbf{p}_k \in \mathbb{S}^{D-1}$ denotes a point on the hypersphere. We first provide the optimization for hyperspherical prototype networks given *a priori* provided prototypes. Then we outline how to find the hyperspherical prototypes in a data-independent manner.

**Loss function and optimization.**  For a single training example $(\mathbf{x}_i, y_i)$, let $\mathbf{z}_i = f_\phi(\mathbf{x}_i)$ denote the $D$-dimensional output vector given a network $f_\phi(\cdot)$. Because we fix the organization of the output space, as opposed to learning it, we propose to train a classification network by minimizing the angle between the output vector and the prototype $\mathbf{p}_{y_i}$ for ground truth label $y_i$, so that the classification loss $\mathcal{L}_c$ to minimize is given as:

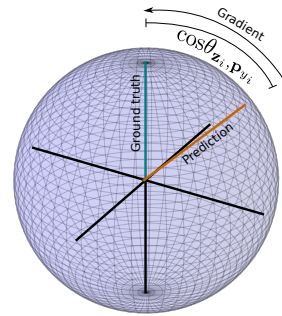

$$\mathcal{L}_c = \sum_{i=1}^N (1 - \cos \theta_{\mathbf{z}_i, \mathbf{p}_{y_i}})^2 = \sum_{i=1}^N (1 - \frac{|\mathbf{z}_i \cdot \mathbf{p}_{y_i}|}{||\mathbf{z}_i||\ ||\mathbf{p}_{y_i}||})^2. \quad (1)$$

The loss function maximizes the cosine similarity between the output vectors of the training examples and their corresponding class prototypes. Figure 2 provides an illustration in output space $\mathbb{S}^2$ for a training example (orange), which moves towards the hyperspherical prototype of its respective class (blue) given the cosine similarity. The higher the cosine similarity, the smaller the squared loss in the above formulation. We note that unlike common classification losses in deep networks, our loss function is only concerned with the map-

Figure 2:  Visualization of hyperspherical prototype network training for classification. Output predictions move towards their prototypes based on angular similarity.

ping from training examples to a pre-defined layout of the output space; neither the space itself nor the prototypes within the output space need to be learned or updated.

Since the class prototypes do not require updating, our network optimization only requires a back-propagation step with respect to the training examples. Let $\theta_i$ be shorthand for $\theta_{\mathbf{z}_i, \mathbf{p}_{y_i}}$. Then the partial derivative of the loss function of Eq. 1 with respect to $\mathbf{z}_i$ is given as:

$$\frac{d}{\mathbf{z}_i}(1 - \cos \theta_i)^2 = 2(1 - \cos \theta_i)\left(\frac{\cos \theta_i \cdot \mathbf{z}_i}{||\mathbf{z}_i||^2} - \frac{\mathbf{p}_{y_i}}{||\mathbf{z}_i||||\mathbf{p}_{y_i}||}\right). \tag{2}$$

The remaining layers in the network are backpropagated in the conventional manner given the error backpropagation of the training examples of Eq. 2. Our optimization aims for angular similarity between outputs and class prototypes. Hence, for a new data point $\tilde{\mathbf{x}}$, prediction is performed by computing the cosine similarity to all class prototypes and we select the class with the highest similarity:

$$c^* = \arg\max_{c \in C} \left(\cos \theta_{f_\phi(\tilde{\mathbf{x}}), \mathbf{p}_c}\right). \tag{3}$$

**Positioning hyperspherical prototypes.** The optimization hinges on the presence of class prototypes that divide the output space prior to learning. Rather than relying on one-hot vectors, which only use the positive portion of the output space and require at least as many dimensions as classes, we incorporate the inductive biases of large margin separation and simplicity. We do so by assigning each class to a single hyperspherical prototype and we distribute the prototypes as uniformly as possible. For $D$ output dimensions and $K$ classes, this amounts to a spherical code problem of optimally separating $K$ classes on the $D$-dimensional unit-hypersphere $\mathbb{S}^{D-1}$ [35]. For $D = 2$, this can be easily solved by splitting the unit-circle $\mathbb{S}^1$ into equal slices, separated by an angle of $\frac{2\pi}{K}$. Then, for each angle $\psi$, the $2D$ coordinates are obtained as $(\cos \psi, \sin \psi)$.

For $D \geq 3$, no such optimal separation algorithm exists. This is known as the Tammes problem [39], for which exact solutions only exist for optimally distributing a handful of points on $\mathbb{S}^2$ and none for $\mathbb{S}^3$ and up [31]. To obtain hyperspherical prototypes for any output dimension and number of classes, we first observe that the optimal set of prototypes, $\mathbf{P}^*$, is the one where the largest cosine similarity between two class prototypes $\mathbf{p}_i, \mathbf{p}_j$ from the set is minimized:

$$\mathbf{P}^* = \arg\min_{\mathbf{P}' \in \mathbb{P}} \left(\max_{(k,l,k \neq l) \in C} \cos \theta_{(\mathbf{p}'_k, \mathbf{p}'_l)}\right). \tag{4}$$

To position hyperspherical prototypes prior to network training, we rely on a gradient descent optimization for the loss function of Eq. 4. In practice, we find that computing all pair-wise cosine similarities and only updating the most similar pair is inefficient. Hence we propose the following optimization:

$$\mathcal{L}_{\text{HP}} = \frac{1}{K} \sum_{i=1}^{K} \max_{j \in C} \mathbf{M}_{ij}, \quad \mathbf{M} = \hat{\mathbf{P}}\hat{\mathbf{P}}^T - 2\mathbf{I}, \quad \text{s.t. } \forall_i \, ||\hat{\mathbf{P}}_i|| = 1, \tag{5}$$

where $\hat{\mathbf{P}} \in \mathbb{R}^{K \times D}$ denotes the current set of hyperspherical prototypes, $\mathbf{I}$ denotes the identity matrix, and $\mathbf{M}$ denotes the pairwise prototype similarities. The subtraction of twice the identity matrix avoids self selection. The loss function minimizes the nearest cosine similarity for each prototype and can be optimized quickly since it is in matrix form. The subtraction of the identity matrix prevents self-selection in the max-pooling. We optimize the loss function by iteratively computing the loss, updating the prototypes, and re-projecting them onto the hypersphere through $\ell_2$ normalization. Compared to uniform sampling methods [14, 30], we explicitly enforce separation. This is because uniform sampling might randomly place prototypes near each other – even though each position on the hypersphere has an equal chance of being selected – which negatively affects the classification.

**Prototypes with privileged information.** So far, no prior knowledge of classes is assumed. Hence all prototypes need to be separated from each other. While separation is vital, semantically unrelated classes should be pushed further away than semantically related classes. To incorporate such privileged information [41] from prior knowledge in the prototype construction, we start from word embedding representations of the class names. We note that the names of the classes generally come for free. To encourage finding hyperspherical prototypes that incorporate semantic information,

we introduce a loss function that measures the alignment between prototype relations and word embedding relations of the classes. We found that a direct alignment impedes separation, since word embedding representations do not fully incorporate a notion of separation. Therefore, we use a ranking-based loss function, which incorporates similarity order instead of direct similarities.

More formally, let $\mathbf{W} = \{\mathbf{w}_1, ..., \mathbf{w}_K\}$ denote the word embeddings of the $K$ classes. Using these embeddings, we define a loss function over all class triplets inspired by RankNet [5]:

$$\mathcal{L}_{\text{PI}} = \frac{1}{|T|} \sum_{(i,j,k) \in T} -\bar{S}_{ijk} \log S_{ijk} - (1 - \bar{S}_{ijk}) \log(1 - S_{ijk}), \qquad (6)$$

where $T$ denotes the set of all class triplets. The ground truth $\bar{S}_{ijk} = [\![\cos \theta_{\mathbf{w}_i, \mathbf{w}_j} \geq \cos \theta_{\mathbf{w}_i, \mathbf{w}_k}]\!]$ states the ranking order of a triplet, with $[\![\cdot]\!]$ the indicator function. The output $S_{ijk} \equiv \frac{e^{o_{ijk}}}{1 + e^{o_{ijk}}}$ denotes the ranking order likelihood, with $o_{ijk} = \cos \theta_{\mathbf{p}_i, \mathbf{p}_j} - \cos \theta_{\mathbf{p}_i, \mathbf{p}_k}$. Intuitively, this loss function optimizes for hyperspherical prototypes to have the same ranking order as the semantic priors. We combine the ranking loss function with the loss function of Eq. 5 by summing the respective losses.

## 2.2 Regression

While existing prototype-based works, e.g. [11, 17, 37], focus exclusively on classification, hyperspherical prototype networks handle regression as well. In a regression setup, we are given $N$ training examples $\{(\mathbf{x}_i, y_i)\}_{i=1}^N$, where $y_i \in \mathbb{R}$ now denotes a real-valued regression value. The upper and lower bounds on the regression task are denoted as $v_u$ and $v_l$ respectively and are typically the maximum and minimum regression values of the training examples. To perform regression with hyperspherical prototypes, training examples should no longer point towards a specific prototype as done in classification. Rather, we maintain two prototypes: $\mathbf{p}_u \in \mathbb{S}^{D-1}$ which denotes the regression upper bound and $\mathbf{p}_l \in \mathbb{S}^{D-1}$ which denotes the lower bound. Their specific direction is irrelevant, as long as the two prototypes are diametrically opposed, i.e. $\cos \theta_{\mathbf{p}_l, \mathbf{p}_u} = -1$. The idea behind hyperspherical prototype regression is to perform an interpolation between the lower and upper prototypes. We propose the following hyperspherical regression loss function:

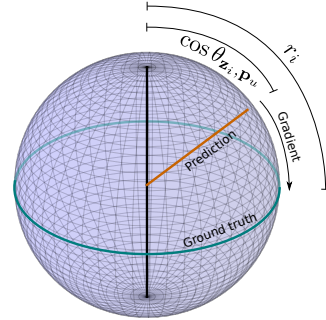

$$\mathcal{L}_{\text{r}} = \sum_{i=1}^N (r_i - \cos \theta_{\mathbf{z}_i, \mathbf{p}_u})^2, \qquad r_i = 2 \cdot \frac{y_i - v_l}{v_u - v_l} - 1. \quad (7)$$

Eq. 7 uses a squared loss function between two values. The first value denotes the ground truth regression value, normalized based on the upper and lower bounds. The second value denotes the cosine similarity between the output vector of the training example and the upper bound prototype. To illustrate the intuition behind the loss function, consider Fig. 3 which shows an artificial training example in output space $\mathbb{S}^2$, with a ground truth regression value $r_i$ of zero. Due to the symmetric nature of the cosine similarity with respect to the upper bound prototype, any output of the training example on the turquoise circle is equally correct. As such, the loss function of Eq. 7 adjusts the angle of the output prediction either away or towards the upper bound prototype, based on the difference between the expected and measured cosine similarity to the upper bound.

Figure 3: Visualization of hyperspherical prototype network training for regression. The output prediction moves angularly towards the turquoise circle, which corresponds to the example's ground truth regression value.

Our approach to regression differs from standard regression, which backpropagate losses on one-dimensional outputs. In the context of our work, this corresponds to an optimization on the line from $\mathbf{p}_l$ to $\mathbf{p}_u$. Our approach generalizes regression to higher dimensional output spaces. While we still interpolate between two points, the ability to project to higher dimensional outputs provides additional degrees of freedom to help the regression optimization. As we will show in the experiments, this generalization results in a better and more robust performance than mean squared error.

Table 1: Accuracy (%) of hyperspherical prototypes compared to baseline prototypes on CIFAR-100 and ImageNet-200 using ResNet-32. Hyperspherical prototypes handle any output dimensionality, unlike one-hot encodings, and obtain the best scores across dimensions and datasets.

| | CIFAR-100 | | | | ImageNet-200 | | | |
|---|---|---|---|---|---|---|---|---|
| *Dimensions* | 10 | 25 | 50 | 100 | 25 | 50 | 100 | 200 |
| One-hot | - | - | - | $62.1 \pm 0.1$ | - | - | - | $33.1 \pm 0.6$ |
| Word2vec | $29.0 \pm 0.0$ | $44.5 \pm 0.5$ | $54.3 \pm 0.1$ | $57.6 \pm 0.6$ | $20.7 \pm 0.4$ | $27.6 \pm 0.3$ | $29.8 \pm 0.3$ | $30.0 \pm 0.4$ |
| This paper | $\mathbf{51.1} \pm 0.7$ | $\mathbf{63.0} \pm 0.1$ | $\mathbf{64.7} \pm 0.2$ | $\mathbf{65.0} \pm 0.3$ | $\mathbf{38.6} \pm 0.2$ | $\mathbf{44.7} \pm 0.2$ | $\mathbf{44.6} \pm 0.0$ | $\mathbf{44.7} \pm 0.3$ |

## 2.3 Joint regression and classification

In hyperspherical prototype networks, classification and regression are optimized in the same manner based on a cosine similarity loss. Thus, both tasks can be optimized not only with the same base network, as is common in multi-task learning [6], but even in the same output space. All that is required is to place the upper and lower polar bounds for regression in opposite direction along one axis. The other axes can then be used to maximally separate the class prototypes for classification. Optimization is as simple as summing the losses of Eq. 1 and 7. Unlike multi-task learning on standard losses for classification and regression, our approach requires no hyperparameters to balance the tasks, as the proposed losses are inherently in the same range and have identical behaviour. This allows us to solve multiple tasks at the same time in the same space without any task-specific tuning.

# 3 Experiments

**Implementation details.** For all our experiments, we use SGD, with a learning rate of 0.01, momentum of 0.9, weight decay of 1e-4, batch size of 128, no gradient clipping, and no pre-training. All networks are trained for 250 epochs, where after 100 and 200 epochs, the learning rate is reduced by one order of magnitude. For data augmentation, we perform random cropping and random horizontal flips. Everything is run with three random seeds and we report the average results with standard deviations. For the hyperspherical prototypes, we run gradient descent with the same settings for 1,000 epochs. The code and prototypes are available at: `https://github.com/psmmettes/hpn`.

## 3.1 Classification

**Evaluating hyperspherical prototypes.** We first evaluate the effect of hyperspherical prototypes with large margin separation on CIFAR-100 and ImageNet-200. CIFAR-100 consists of 60,000 images of size 32x32 from 100 classes. ImageNet-200 is a subset of ImageNet, consisting of 110,000 images of size 64x64 from 200 classes [18]. For both datasets, 10,000 examples are used for testing. ImageNet-200 provides a challenging and diverse classification task, while still being compact enough to enable broad experimentation across multiple network architectures, output dimensions, and hyperspherical prototypes. We compare to two baselines. The first consists of one-hot vectors on the $C$-dimensional simplex for $C$ classes, as proposed in [2, 7]. The second baseline consists of word2vec vectors [27] for each class based on their name, which also use the cosine similarity to compare outputs, akin to our setting. For both baselines, we only alter the prototypes compared to our approach. The network architecture, loss function, and hyperparameters are identical. This experiment is performed for four different numbers of output dimensions.

The results with a ResNet-32 network [13] are shown in Table 1. For both CIFAR-100 and ImageNet-200, the hyperspherical prototypes obtain the highest scores when the output size is equal to the number of classes. The baseline with one-hot vectors can not handle fewer output dimensions. Our approach can, and maintains accuracy when removing three quarters of the output space. For CIFAR-100, the hyperspherical prototypes perform 7.4 percent points better than the baseline with word2vec prototypes. On ImageNet-200, the behavior is similar. When using even fewer output dimensions, the relative accuracy of our approach increases further. These results show that hyperspherical prototype networks can handle any output dimensionality and outperform prototype alternatives. We have performed the same experiment with DenseNet-121 [16] in the supplementary materials, where we observe the same trends; we can trim up to 75 percent of the output space while maintaining accuracy, outperforming baseline prototypes.

In Table 2, we have quantified prototype separation of the three approaches with 100 output dimensions on CIFAR-100. We calculate the min (cosine distance of closest pair), mean (average pair-wise cosine distance), and max (cosine distance of furthest pair) separation. Our approach obtains the highest mean and maximum separation, indicating the importance of pushing many classes beyond orthogonality. One-hot prototypes do not push beyond orthogonality, while word2vec prototypes have a low minimum separation, which induces confusion for semantically related classes. These limitations of the baselines are reflected in the classification results.

Table 2: Separation stats.

| | Separation ↑ | | |
| | min | mean | max |
| --- | --- | --- | --- |
| One-hot | **1.00** | 1.00 | 1.00 |
| Word2vec | 0.26 | 1.01 | 1.32 |
| This paper | 0.95 | **1.01** | **1.39** |

**Prototypes with privileged information.** Next, we investigate the effect of incorporating privileged information when obtaining hyperspherical prototypes for classification. We perform this experiment on CIFAR-100 with ResNet-32 using 3, 5, 10, and 25 output dimensions. The results in Table 3 show that incorporating privileged information in the prototype construction is beneficial for classification. This holds especially when output spaces are small. When using only 5 output dimensions, we obtain an accuracy of $37.0 \pm 0.8$, compared to $28.7 \pm 0.4$, a considerable improvement. The same holds when the number of dimensions is larger than the number of classes. Separation optimization becomes more difficult, but privileged information alleviates this problem.

We also find that the test convergence rate over training epochs is higher with privileged information. We highlight this in the supplementary materials. Privileged information results in a faster convergence, which we attribute to the semantic structure in the output space. We conclude that privileged information improves classification, especially when the number of output dimensions does not match with the number of classes.

Table 3: Classification accuracy (%) on CIFAR-100 using ResNet-32 with and without privileged information in the prototype construction. Privileged information aids classification, especially for small outputs.

| | **CIFAR-100** | | | | | |
| *Dimensions* | 3 | 5 | 10 | 25 | 100 | 200 |
| --- | --- | --- | --- | --- | --- | --- |
| Hyperspherical prototypes | $5.5 \pm 0.3$ | $28.7 \pm 0.4$ | $51.1 \pm 0.7$ | $63.0 \pm 0.1$ | **65.0** $\pm 0.3$ | $63.7 \pm 0.4$ |
| w/ privileged info | **11.5** $\pm 0.4$ | **37.0** $\pm 0.8$ | **57.0** $\pm 0.6$ | **64.0** $\pm 0.2$ | $63.8 \pm 0.1$ | **64.7** $\pm 0.1$ |

**Comparison to other prototype networks.** Third, we consider networks where prototypes are defined as the class means using the Euclidean distance, e.g. [11, 17, 37, 45]. We compare to Deep NCM of Guerriero et al. [11], since it can handle any number of training examples and any output dimensionality, akin to our approach. We follow [11] and report on CIFAR-100. We run the baseline provided by the authors with the same hyperparameter settings and network architecture as used in our approach. We report all their approaches for computing prototype: mean condensation, mean decay, and online mean updates.

In Fig. 4, we provide the test accuracy as a function of the training epochs on CIFAR-100. Overall, our approach provides multiple benefits over Deep NCM [11]. First, the convergence of hyperspherical prototype networks is faster and reaches better results than the baselines. Second, the test accuracy of hyperspherical prototype networks changes smoother over iterations. The test accuracy gradually improves over the training epochs and quickly converges, while the test accuracy of the baseline behaves more erratic between training epochs. Third, the optimization of hyperspherical prototype networks is computationally easier and more efficient. After a feed forward step through the network, each training example only needs to compute the cosine similarity with respect to their class prototypes. The baseline needs to compute a distance to *all* classes, followed by a softmax. Furthermore, the class prototypes require constant updating, while our prototypes remain fixed. Lastly, compared to other prototype-based networks, hyperspherical prototype networks are easier to implement and require only a few lines of code given pre-computed prototypes.

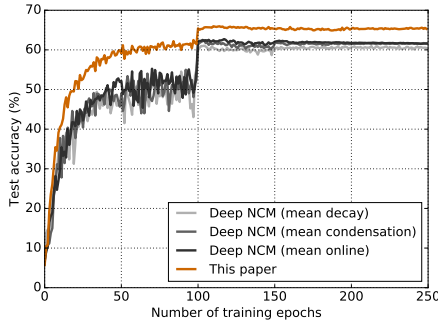

Figure 4: Comparison to [11]. Our approach outperforms the baseline across all their settings with the same network hyperparameters and architecture.

**Comparison to softmax cross-entropy.** Fourth, we compare to standard softmax cross-entropy classification. For fair comparison, we use the same number of output dimensions as classes for hyperspherical prototype networks, although we are not restricted to this setup, while softmax cross-entropy is. We report results in Table 4. First, when examples per class are large and evenly distributed, as on CIFAR-100, we obtain similar scores. In settings with few or uneven samples, our approach is preferred. To highlight this ability, we have altered the train and test class distribution on CIFAR-100, where we linearly increase the number of training examples for each class, from 2 up to 200. For such a distribution, we outperform softmax cross-entropy. In our approach, all classes have a roughly equal portion of the output space, while this is not to for softmax cross-entropy in uneven settings [20]. We have also performed an experiment on CUB Birds 200-2011 [42], a dataset of 200 bird species, 5,994 training, and 5,794 test examples, i.e. a low number of exam-
ples per class. On this dataset, we perform better than softmax cross-entropy under identical networks and hyperparameters ($47.3 \pm 0.1$ vs $43.0 \pm 0.6$). Lastly, we have compared our approach to a softmax cross-entropy baseline which learns a cosine similarity using all class prototypes. This baseline obtains an accuracy of $55.5 \pm 0.2$ on CIFAR-100, not competitive with standard softmax cross-entropy and our approach. We conclude that we are comparable to softmax cross-entropy for sufficient examples and preferred when examples per class are unevenly distributed or scarce.

Table 4: Accuracy (%) for our approach compared to softmax cross-entropy. When examples per class are scarce or uneven, our approach is preferred.

|  | **CIFAR-100** | | **CUB-200** |
| --- | --- | --- | --- |
| *ex / class* | 500 | 2 to 200 | $\sim$30 |
| Softmax CE | $64.4 \pm 0.4$ | $44.2 \pm 0.0$ | $43.1 \pm 0.6$ |
| This paper | $65.0 \pm 0.3$ | $\mathbf{46.4} \pm 0.0$ | $\mathbf{47.3} \pm 0.1$ |

## 3.2 Regression

Next, we evaluate regression on hyperspheres. We do so on the task of predicting the creation year of paintings from the $20^{th}$ century, as available in OmniArt [38]. This results in a dataset of 15,000 training and 8,353 test examples. We use ResNet-32, trained akin to the classification setup. Mean Absolute Error is used for evaluation. We compare to a squared loss regression baseline, where we normalize and 0-1 clamp the outputs using the upper and lower bounds for a fair comparison. We create baseline variants where the output layer has more dimensions, with an additional layer to a real output to ensure at least as many parameters as our approach.

Table 5 shows the regression results of our approach compared to the baseline. We investigate both $\mathbb{S}^1$ and $\mathbb{S}^2$ as outputs. When using a learning rate of 1e-2, akin to classification, our approach obtains an MAE of 84.4 ($\mathbb{S}^1$) and 82.9 ($\mathbb{S}^2$). The baseline yields an error rate of respectively 210.7 and 339.9, which we found was due to exploding gradients. Therefore, we also employed a learning rate of 1e-3, re-

Table 5: Mean absolute error rates for creation year on artistic images in Omniart. Our approach obtains the best results and is robust to learning rate settings.

|  | **Omniart** | | | |
| --- | --- | --- | --- | --- |
| *Output space* | $\mathbb{S}^1$ | | $\mathbb{S}^2$ | |
| *Learning rate* | 1e-2 | 1e-3 | 1e-2 | 1e-3 |
| MSE | $210.7 \pm 140.1$ | $110.3 \pm 0.8$ | $339.9 \pm 0.0$ | $109.9 \pm 0.5$ |
| This paper | $\mathbf{84.4} \pm 10.7$ | $\mathbf{76.3} \pm 5.6$ | $\mathbf{82.9} \pm 1.9$ | $\mathbf{73.2} \pm 0.6$ |

sulting in an MAE of 76.3 ($\mathbb{S}^1$) and 73.2 ($\mathbb{S}^2$) for our approach, compared to 110.0 and 109.9 for the baseline. While the baseline performs better for this setting, our results also improve. We conclude that hyperspherical prototype networks are both robust and effective for regression.

## 3.3 Joint regression and classification

**Rotated MNIST.** For a qualitative analysis of the joint optimization we use MNIST. We classify the digits and regress on their rotation. We use the digits 2, 3, 4, 5, and 7 and apply a random rotation between 0 and 180 degrees to each example. The other digits are not of interest given the rotational range. We employ $\mathbb{S}^2$ as output, where the classes are separated along the $(x, y)$-plane and the rotations are projected along the $z$-axis. A simple network is used with two convolutional and two fully connected layers. Fig. 5 shows how in the same space, both image rotations and classes can be modeled. Along the $z$-axis, images are gradually rotated, while the $(x, y)$-plane is split into maximally separated slices representing the classes. This qualitative result shows both tasks can be modeled jointly in the same space.

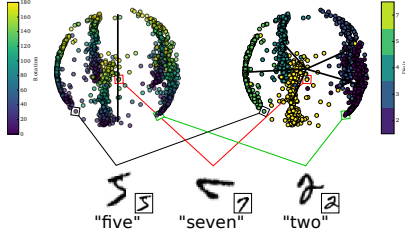

Figure 5: Joint regression and classification on rotated MNIST. Left: colored by rotation (z-axis). Right: colored by class assignment (xy-plane).

Table 6: Joint creation year and art style prediction on OmniArt. We are preferred over the multi-task baseline regardless of any tuning of the task weight, highlighting the effectiveness and robustness of our approach.

| Task weight | 0.01 | 0.10 | 0.25 | 0.50 | 0.90 |
|---|---|---|---|---|---|
| Creation year (MAE ↓) | | | | | |
| MTL baseline | 262.7 | 344.5 | 348.5 | 354.7 | 352.3 |
| This paper | 65.2 | 64.6 | **64.1** | 68.3 | 83.6 |
| Art style (acc ↑) | | | | | |
| MTL baseline | 44.6 | 47.9 | 49.5 | 47.2 | 47.1 |
| This paper | 46.6 | 51.2 | **54.5** | 52.6 | 51.4 |

**Predicting creation year and art style.** Finally, we focus on jointly regressing the creation year and classifying the art style on OmniArt. There are in total 46 art styles, denoting the school of the artworks, e.g. the *Dutch* and *French* schools. We use a ResNet-32 with the same settings as above (learning rate is set to 1e-3). We compare to a multi-task baseline, which uses the same network and settings, but with squared loss for regression and softmax cross-entropy for classification. Since this baseline requires task weighting, we compare both across various relative weights between the regression and classification branches. The results are shown in Table 6. The weights listed in the table denote the weight assigned to the regression branch, with one minus the weight for the classification branch. We make two observations. First, we outperform the multi-task baseline across weight settings, highlighting our ability to learn multiple tasks simultaneously in the same shared space. Second, we find that the creation year error is lower than reported in the regression experiment, indicating that additional information from art style benefits the creation year task. We conclude that hyperspherical prototype networks are effective for learning multiple tasks in the same space, with no need for hyperparameters to weight the individual tasks.

# 4 Related work

Our proposal relates to prototype-based networks, which have gained traction under names as proxies [29], means [11], prototypical concepts [17], and prototypes [9, 19, 33, 36, 37]. In general, these works adhere to the Nearest Mean Classifier paradigm [26] by assigning training examples to a vector in the output space of the network, which is defined to be the mean vector of the training examples. A few works have also investigated multiple prototypes per class [1, 29, 46]. Prototype-based networks result in a simple output layout [45] and generalize quickly to new classes [11, 37, 46].

While promising, the training of these prototype networks faces a chicken-or-egg dilemma. Training examples are mapped to class prototypes, while class prototypes are defined as the mean of the training examples. Because the projection from input to output changes continuously during network training, the true location of the prototypes changes with each mini-batch update. Obtaining the true location of the prototypes is expensive, as it requires a pass over the complete dataset. As such, prototype networks either focus on the few-shot regime [4, 37], or on approximating the prototypes, e.g. by alternating the example mapping and prototype learning [12] or by updating the prototypes online as a function of the mini-batches [11]. We bypass the prototype learning altogether by structuring the output space prior to training. By defining prototypes as points on the hypersphere, we are able to separate them with large margins *a priori*. The network optimization simplifies to minimizing a cosine distance between training examples and their corresponding prototype, alleviating the need to continuously obtain and learn prototypes. We also generalize beyond classification to regression using the same optimization and loss function.

The work of Perrot and Habard [34] relates to our approach since they also use pre-defined prototypes. They do so in Euclidean space for metric learning only, while we employ hyperspherical prototypes for classification and regression in deep networks. Bojanowski and Joulin [3] showed that unsupervised learning is possible through projections to random prototypes on the unit hypersphere and updating prototype assignments. We also investigate hyperspherical prototypes, but do so in a supervised setting, without the need to perform any prototype updating. In the process, we are encouraged by Hoffer et al. [15], who show the potential of fixed output spaces. Several works have investigated prior and semantic knowledge in hyperbolic spaces [8, 32, 40]. We show how to embed prior knowledge in hyspheric spaces and use it for recognition tasks. Liu et al. [20] propose a large margin angular

separation of class vectors through a regularization in a softmax-based deep network. We fix highly separated prototypes prior to learning, rather than steering them during training, while enabling the use of prototypes in regression.

Several works have investigated the merit of optimizing based on angles over distances in deep networks. Liu et al. [23], for example, improve the separation in softmax cross-entropy by increasing the angular margin between classes. In similar fashion, several works project network outputs to the hypersphere for classification through $\ell_2$ normalization, which forces softmax cross-entropy to optimize for angular separation [12, 21, 22, 43, 44, 47]. Gidaris and Komodakis [10] show that using cosine similarity in the output helps generalization to new classes. The potential of angular similarities has also been investigated in other layers of deep networks [24, 25]. We also focus on angular separation in deep networks, but do so from the prototype perspective.

## 5  Conclusions

This paper proposes hyperspherical prototype networks for classification and regression. The key insight is that class prototypes should not be a function of the training examples, as is currently the default, because it creates a chicken-or-egg dilemma during training. Indeed, when network weights are altered for training examples to move towards class prototypes in the output space, the class prototype locations alter too. We propose to treat the output space as a hypersphere, which enables us to distribute prototypes with large margin separation without the need for any training data and specification prior to learning. Due to the general nature of hyperspherical prototype networks, we introduce extensions to deal with privileged information about class semantics, continuous output values, and joint task optimization in one and the same output space. Empirically, we have learned that hyperspherical prototypes are effective, fast to train, and easy to implement, resulting in flexible deep networks that handle classification and regression tasks in compact output spaces. Potential future work for hyperspherical prototype networks includes incremental learning and open set learning, where the number of classes in the vocabulary is not fixed, requiring iterative updates of prototype locations.

## Acknowledgements

The authors thank Gjorgji Strezoski and Thomas Mensink for help with datasets and experimentation.

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
