[Supplementary Material · hyperspherical_prototype_learning_neurips19supp.pdf]

# Hyperspherical Prototype Networks

## Supplementary Materials

## 1 Evaluating hyperspherical prototypes on DenseNet-121

Table 1 provides an overview of the evaluation of hyperspherical prototypes with a DenseNet-121 network architecture. The setup is identical to the setup of the first experiment of the main paper, where ResNet-32 was used as architecture. We reach the same conclusions as for the experiment using ResNet-32; our hyperspherical prototypes enable a large reduction in output dimensionality while maintaining performance, outperforming the baseline prototypes on both datasets.

| | CIFAR-100 | | | ImageNet-200 | | |
|---|---|---|---|---|---|---|
| *Dimensions* | 10 | 25 | 100 | 25 | 50 | 200 |
| One-hot | - | - | 72.2 | - | - | 60.4 |
| Word2vec | 36.6 | 61.0 | 71.9 | 35.9 | 46.0 | 56.6 |
| This paper | **60.5** | **69.1** | **73.4** | **56.0** | **60.6** | **62.6** |

Table 1: Accuracy (%) of our hyperspherical prototypes compared to baseline prototypes using DenseNet-121. Akin to the experiments on ResNet-32, hyperspherical prototypes can handle any output dimensionality, unlike one-hot encodings, and obtain the best scores across datasets.

## 2 Accuracy per epoch for prototypes with privileged information

In Figure 1, we show the classification accuracy on CIFAR-100 using ResNet-32 as a function of the number of training epochs. We compare our approach both with and without the use of privileged information about class semantics, using 3, 5, 10, and 25 output dimensions. The Figure shows that the use of privileged information not only leads to better results, but also faster convergence. We attribute this to the semantic structure of the output space, which leads to a smoother optimization. For larger output spaces, the importance of privileged information is less vital, since there is more space for class prototypes to be separated from every other class, regardless of semantic structure.

(a) 3 dimensions.

(b) 5 dimensions.

(c) 10 dimensions.

(d) 25 dimensions.

Figure 1: Classification accuracy (%) on CIFAR-100 using ResNet-32 for our approach with and without privileged information. We show the performance interpolated based on scores for every 10 epochs. Incorporating privileged information results in faster convergence and better overall results, especially in small output spaces.