[Reviews · NeurIPS 2019]

Reviewer 1



The paper proposes a new read-out layer, hyperspherical output space, for both classification and regression. Before starting training, the last fully connected (FC) layer of the network is designed by fixed anchor points (not trained) on the hyperspherical space, called prototypes. The network outputs the inner product values between the activation from the penultimate layer and the prototypes, and used as a logit for classification or bounded regression value for regression. Since finding uniformly distributed anchor points in general D>2 is not trivial, the authors proposed a simple gradient descent based method. Then, the obtained points are fixed and used for the prototypes of the network. The paper demonstrates the application of this prototype for N-way classification with N number of prototypes and bounded value regression with 2 prototypes located at the random poles in a unified way. The paper is well-written, and the reasonable experiments are provided to validate the proposed idea (it could have been stronger though). Especially, I like the regression idea. I also have personally experienced the design limitation of the conventional deep regression (1-dimensional projection by the final FC layer). I found that this work presents a good alternative that has a better-relaxed lifting mechanism for complex regression. Also, if the prototypes are randomly assigned to each class, inherent taxonomy property of the class would be ignored. The authors did a good try to compensate for the non-learnable limitation of the prototype by borrowing privileged external information. Although it ends up with sub-optimal, it would be effective in some cases. - On the prototypes generation method The authors proposed a gradient descent approach which is also an approximate method. But there has been long-standing research on uniform sampling points on N-dim. hypersphere (called hypersphere point picking problem) [C1,C2]. Efficient uniform random sampling methods on N-dim. hypersphere would be much simpler and effective alternatives. How much the network performance is tolerant to the uniformity of the prototypes for the classification task? This would be an important analysis required to understand the behavior of the proposed algorithm. [C1] An Efficient Method for Generating Uniformly Distributed Points on the Surface of an n-Dimensional Sphere, ACM Comm., 1959. [C2] A note on a method for generating points uniformly on n-dimensional spheres, ACM Comm., 1959. - Tables 1 and 2 have inconsistent dimensions which prevent from directly comparing both experiments. At least, D-100 would be helpful to understand the network behavior. - Balancing parameter test (L163-167) Although it is working plausibly, setting the balance parameters to 1 would never be optimal setup. Thus, introducing a balance parameter and analyzing its effect would provide a good sense of the behavior under the hood. L310 (no weight fo balance the losses for the multi-task baseline); this seems an unfair comparison because there must be significant differences on statics of scale between the logits for classification (softmax cross-entropy) and the regression output for the regression. Thus, using the weights equal to 1 for both the baseline and the proposed method is entirely unfair. Since the proposed design to have an equal range of outputs with the same class of loss could be viewed as a loss balancing by design, it would be an appropriate comparison if the balance parameters are introduced and tuned to be the best for respective algorithms. I guess, since the proposed method would definitely favor benefits from the equal range over the baseline, nonetheless it would be a more valid experiment setup. The 2nd baseline used word2vec embedding as a prototype. Word2vec embedding space tends to have Euclidean space rather than hyperspherical space. This raises several questions. What loss is used, and what structure the output layer has? Is the baseline just same with the proposed design with replacing the prototype to be word2vec embeddings? Since word2vec embeddings tend to lie on Euclidean space, it may not make sense to use inner product or cross-entropy for the baseline, which is incompatible with word2vec as prototypes. - L184-185: Another unclear part is on the "one-hot" baseline in Table 1 and Table 1 in the supplementary. The sentence is confusing; it said something about softmax cross-entropy but appears that the authors used the same cosine loss with one-hot (axis-aligned) prototypes for the baseline. Is this correct? It seems this case, because the authors provide a separate comparison in the paragraph "Comparison to softmax cross-entropy" (Table 3). If this is the case, then it would have been clearer to use the same prototype dimensions across Tables 1, 2 and 3, so that those experiments are comparable each other. Also, one concern raised by this is the baseline is not strong enough. Why the authors didn't directly compare with the standard classification layer in fair settings? Tables 1, 2 and 3 have all subtle difference which prevents from directly comparison. This concern reduces my rating. The proposed method apparently has cases that there would be no gain by taking the proposed method. For instance, if the prototype is designed to have one-hot vectors for each class, then this works like selecting a dimension to obtain a pseudo probability value from the activation of the penultimate layer. This case could be detrimental because this is likely to reduce the network capacity as much as the capacity of the one layer that can be obtained from the well-posed prototype. Also, it is exactly applicable to the case of the regression. Thus, the success and improvement of the proposed method may be related to the dimensionality of the prototype and the direction of the prototype (distribution or sparsity and so on). This would be valuable for improving the proposed method by designing to avoid such cases in practice.

Reviewer 2



Strengths – The paper presents a novel and well-motivated approach that is crisply explained, with significant experimental results to back it up. Clean and well-documented code has been provided and the results should be easily reproducible. – The wide variety of applications covered demonstrate its versatility. In particular, the data-independent optimization, ability to bake in priors, computational savings by trimming output dimensionality, improvements over (de-facto) softmax classification in the presence of class imbalance, and suitability for multitask learning without loss weighting are strong wins. – The paper is well-written and very easy to follow. The related section does well to clearly identify and present threads in prior work. Weaknesses / Questions – What is the performance of the multitask baseline (Table 5) with appropriate loss weighting? – L95: “We first observe that the optimal set of prototypes .. one with largest cosine similarity”: How has this observation been made? Is this conjecture, or based on prior work? – What is the computational efficiency of training hyperspherical networks as compared to prior work? In particular, the estimation of prototypes (L175: gradient descent .. 1000 epochs) seems like a particularly expensive step. – Prototypical networks were initially proposed as a few-shot learning approach, with the ability to add additional classes at test time with a single forward pass. One limitation of the proposed approach is not being suitable for such a setup without retraining from scratch. A similar argument would apply for extending to open set classification, as the approach divides up the output space into as many regions as classes. While the paper does not claim this approach extends to either of these settings, have the authors thought about whether / how such an extension would be possible? – Minor: – Have the authors investigated the reason for the sharp spike in performance for both NCM and the proposed approach at 100 epochs as shown in Figure 4? – It is not clear from Figure 1b how the approach is suitable for regression, and a more detailed caption would possibly help. – Typo: L85: as many dimensions *as* classes

Reviewer 3



First and maybe most importantly, how to reasonably assign these class prototypes. The paper makes it an optimization problem which resembles Tammes problem. It can be sometimes problematic, since Tammes problem in high dimensions is highly non-convex and the optimal solution can be be obtained or even evaluated. Using gradient-based optimization is okay but far away from satisfactory, and most importantly, you can not evaluate whether you obtain a good local minima or not. Besides this, the semantic separation and maximal separation are not necessarily the same. As you mentioned in your paper, car and tighter can be similar than can and bulldozer. But how to incorporate this prior remains a huge challenge, because it involves the granularity of the classes. Naively assign class prototype on hypersphere could violate the granularity of the classes. The paper considers a very simple way to alleviate this problem by introducing a difference loss between word2vec order and the prototype order, which makes senses to me. However, it is still far from satisfactory, especially when you have thousands of classes. The class prototype assignment in high dimensions could lead to a huge problem. From my own experience, I have manually assigned CNN classifiers which have maximal inter-class distance, and then train the CNN features with these classifiers being fixed the whole time. My results show that it can be really difficult for the network to converge. Maybe adding a privileged information like the authors did could potentially help, but I am not very sure about it. Second, I like the idea of using hypersphere as the output space despite the possible technical difficulties. I have a minor concern for the classification loss. The classification loss takes the form of a least square minimization which can essentially viewed as a regression task. What if using the softmax cross-entropy loss instead? Will it be better or worse? I am quite curious about the performance

[Author Response · NeurIPS 2019]

We thank all the reviewers (**R1,R2,R3**) for their feedback and suggestions.

**Multi-task loss balancing (R1,R2).** We have per-
formed loss balancing with five different weights $t$

Table A: Multi-task comparison across task weights.

| Task weight | 0.01 | 0.10 | 0.25 | 0.50 | 0.75 | 0.90 |
|---|---|---|---|---|---|---|
| Creation year (MAE ↓) | | | | | | |
| MTL baseline | 262.7 | 344.5 | 348.5 | 354.7 | 356.3 | 352.3 |
| This paper | 65.2 | 64.6 | **64.1** | 68.3 | 77.5 | 83.6 |
| Art style (acc ↑) | | | | | | |
| MTL baseline | 44.6 | 47.9 | 49.5 | 47.2 | 47.7 | 47.1 |
| This paper | 46.6 | 51.2 | **54.5** | 52.6 | 52.5 | 51.4 |

in the multi-task loss $\mathcal{L}_m = t \cdot \mathcal{L}_c + (1-t) \cdot \mathcal{L}_r$ for
the classification and regression losses. The results
on OmniArt are reported in Table A. Our proposal
is robust to the weight value, tuning the task weight
is not vital. We obtain a moderate gain for both clas-
sification and regression with a weight of $t = 0.25$.
For the multi-task baseline, emphasizing regression
reduces the regression error, as the gradient magnitude of the regression loss is much lower than the one for the
classification loss. However, this is not paired with an increase in classification accuracy. Across all weights, our
proposal is preferred over the baseline. We will add the suggested experiment to Section 3.3. Thank you.

**Softmax cross-entropy loss (R3).** We tried the suggested softmax cross-entropy loss on the hyperspherical similarities
to all class prototypes. On CIFAR-100, we obtain an accuracy of $55.5 \pm 0.2$, compared to $65.0 \pm 0.3$ with our loss
and $64.4 \pm 0.4$ with standard softmax cross-entropy, all with the same output dimensions and network settings. The
softmax cross-entropy variant obtains a lower accuracy and is computationally more expensive, as it needs a similarity
to all classes to compute the loss. The proposed loss requires a similarity to only one class during training. We will add
the discussion to Section 3.1.

**Hyperspherical separation and effect on classification (R1,R2).** To **R2**,
the statement on L95 regarding optimal separation is based on [33]. We

Table B: Separation vs. classification.

| | Separation ↑ | | | acc. |
|---|---|---|---|---|
| | min | mean | max | |
| One-hot | **1.00** | 1.00 | 1.00 | 62.1 |
| Word2vec | 0.26 | 1.01 | 1.32 | 57.6 |
| This paper | 0.95 | **1.01** | **1.39** | **65.0** |

will add the reference. Obtaining hyperspherical prototypes is fast: with 100
classes in 100 dimensions and 1,000 epochs, optimization takes only 4 seconds
on a single 1080TI. Following **R1**'s suggestion, we have quantified the relation
between separation and classification accuracy for the two hyperspherical
baselines and our proposal with 100 output dimensions on CIFAR-100. We
calculate the min (cosine distance of closest pair), mean (average pair-wise
cosine distance), and max (cosine distance of furthest pair) separation. The results are shown in Table B. Our proposal
obtains the highest maximum separation, indicating the importance of pushing many classes beyond orthogonality.
One-hot prototypes do not push beyond orthogonality, while word2vec prototypes have a low minimum separation,
which induces class confusion. To **R1**, we find degenerate cases on CIFAR-100 with three dimensions; the prototypes
of multiple classes are overlapping. This case can either be solved by incorporating privileged information, or by using
(at least two) more dimensions for the hypersphere (Table 2 of paper). We will add the discussions to Section 3.1.

**More dimensions than classes (R1,R3).** We have performed an additional experiment on CIFAR-100 with 200 output
dimensions. We obtain an accuracy of $63.7 \pm 0.4$ (without privileged information) and $64.7 \pm 0.1$ (with privileged
information), compared to $65.0 \pm 0.3$ for 100 dimensions. As foreseen by **R3**, the additional dimensions makes
optimization more difficult, but privileged information alleviates this problem. We will insert the experiments with
additional dimensions in Tables 1 and 2 and incorporate the discussion.

**Extension to few-shot and open set classification (R2).** In standard few-shot, the total number of classes is known
[35], allowing us to place all classes on the hypersphere *a priori*. For open set classification, we can avoid retraining
from scratch by extending the prototype separation. For a new class, we place it on the hypersphere (e.g. with privileged
information) and then push all prototypes near the new one away. We will add these possibilities to the conclusions.

**Clarifying one-hot and word2vec baselines (R1).** For the word2vec and one-hot baselines (Tables 1 and 2), we only
replace the prototypes, the loss and optimization remains the same. We will remove L184-185 to avoid confusion on the
one-hot baseline. The word2vec baseline relies on word vectors of class names as the prototypes, which is feasible
because the word vectors use the cosine similarity as distance metric [35], akin to hyperspherical prototype networks.
The comparison to standard classification with softmax cross-entropy (Table 3) uses the same settings and dimensions,
only the loss is altered. We believe this setup allows for a direct and fair comparison, but the description of baselines
and experiments could surely be improved. We will do so and align the dimensions across Tables 1-3 to allow for
cross-experiment comparisons.

We thank **R1** for the references. Compared to uniform sampling, we explicitly enforce maximum separation, because
uniform sampling might randomly place prototypes near each other, which negatively affects the classification (Table B).
We will cite and discuss the papers in Section 2.1. To **R2**, the sharp spike in Figure 4 is due to a learning rate decay after
100 epochs. We will update the caption of Figure 1b to increase clarity and remove the typo. We thank the reviewers.

[Meta-Review · NeurIPS 2019]

The reviewers all agree that this paper presents an interesting alternative output layer for classification and regression networks. The reviewers were especially satisfied with the additional results and content of the rebuttal, so please be sure to include this in the final draft.